# Model-Based Optimization of an LLC-Resonant DC-DC Converter

**Nikolay Hinov [1,*]** , **Bogdan Gilev [2] and Tsveti Hranov [1]**

[1]  Faculty Electronic Engineering and Technology, Technical University of Sofia, Sofia 1000, Bulgaria
[2]  Faculty of Applied Mathematics and Informatics, Technical University of Sofia, Sofia 1000, Bulgaria
*  Correspondence: hinov@tu-sofia.bg; Tel.: +359-2-965-2569

**Abstract:** The study presented in the paper is to guarantee the performance of the LLC DC-DC converter using model-based optimization. The primary scope of the study is to maintain the output parameters regardless of the variation of the values of the circuit elements. In engineering practice, it is known that any schematic element cannot be reproduced with an absolute accuracy of features. In addition, its main parameters change during operation due to changes in operating temperature, aging, operating modes and so on. Optimization procedures are a tool for finding the most appropriate values for circuit elements, with selected constraints, target functions and operating modes. In electronic converters, these are most often: minimal loss, maximum efficiency, the critical-aperiodic transition process, realization of certain dynamics, appropriate modes of operation and so on. The results obtained show that using the proposed approach produces more robustness to disturbances and tolerances, with improved dynamics and faster transient processes. On the other hand, the value of the circuit elements is smaller, and reliable operation of the protection and automatic regulation systems is achieved.

**Keywords:** LLC DC-DC converter; mathematical optimizations; modeling; guaranteed parameters; tolerance analysis

## 1. Introduction

Optimization is one of the most commonly used tools in research. Its primary goal is to get the best result using a specified target function with respect to given constraints. In electronic converters, optimization procedures are most often related to: minimal loss, maximum efficiency, the critical-aperiodic transition process, realization of a certain dynamics, appropriate modes of operation and so on.

Each schematic element possesses nonlinear properties to a certain degree (i.e., its characteristics are influenced by voltage and current loads). In this sense, the problem of guaranteeing the performance of a power electronic device is formulated as follows. The variations of the circuit elements characteristics are obtained through formulated areas reflecting changes in parameters and the operating modes of the device. For example, with resonant inductance tolerance values within ±20% of the nominal value, and resonant capacitance values in the vicinity of ±10%, an examination of the changes in output voltage, resonant current and voltage across the resonant capacitor define an area of operating modes for the LLC converter.

The main objective of this study is to evaluate the influence of the tolerances of a mathematical model of the device by varying parts of the LLC DC-DC converter circuit element operating modes, and performing various optimization procedures aimed at guaranteeing its performance.

## 2. Models, Procedures and Methods of Study

It is well known that the operating modes of resonant DC converters are strongly dependent on changes in the values of inverter circuit elements [1–4]. At the same time, most LLC DC-DC converter engineering techniques have been developed with an assumption that the converter is working in a steady-state mode, and that currents and voltages are sinusoidal. In such a situation, automatic regulation systems are necessary to guarantee the performance of indicators. When the converter parameters are going out of bounds due to the influence of disturbances (manufacturing tolerances, non-linear effects, aging) outside the assumptions made at the design stage, it cannot be guaranteed that the control system will be able to return the converter to the nominal mode.

One of the possible approaches to guaranteeing LLC converter performance is the use of optimization through the use of mathematical methods to perform certain automated procedures. The specific software implementation was performed using models, optimization criteria, range and steps of varying the tolerances of the circuit elements by different criteria.

A number of approaches have been developed to optimize power electronic devices [5–8]. Typically, these devices are based on sophisticated mathematical methods employed by engineers and designers, who actually design the devices, and that are difficult to reproduce. Optimizations are focused primarily on loss reduction and efficiency gain [7,8], and not applied to improve dynamics or guarantee a device's performance. The necessary dynamical characteristics are achieved mainly with the controller and the use of more complex control methods [9,10] instead of the optimization of other parameters of the circuit.

The improvement of the dynamics is usually connected with the synthesis of a controller, usage of more modern control algorithms such as neural networks, predictive control and so on. This is also demonstrated in [11,12], which are dedicated to improving the dynamics of DC-DC power electronic converters.

In this study, a model-based optimization is proposed that includes a model of the device, a reference curve or several reference curves and the use of standard functions embedded in MATLAB. Optimization procedures could be carried out as additional design steps after the main values are calculated in the main design stage. The advantage of model-based optimization is that it is applicable practically to all electronic devices, not strictly resonant converters.

### 2.1. Mathematical Model

The circuit of the resonant LLC DC-DC converter is given in Figure 1. It is composed of the electronic switches VS1 and VS2, resonant inductor $L_r$ and capacitor $C_r$ (split in two equal parts $C_{r1}$ and $C_{r2}$). In order to obtain DC voltage, the output is used a rectifier. In the output circuit are connected the output filter capacitor $C_0$ and load $R$. The load is coupled via the resonant transformer $Tr$. In order to obtain greater efficiency, the circuit could be used a synchronous rectifier.

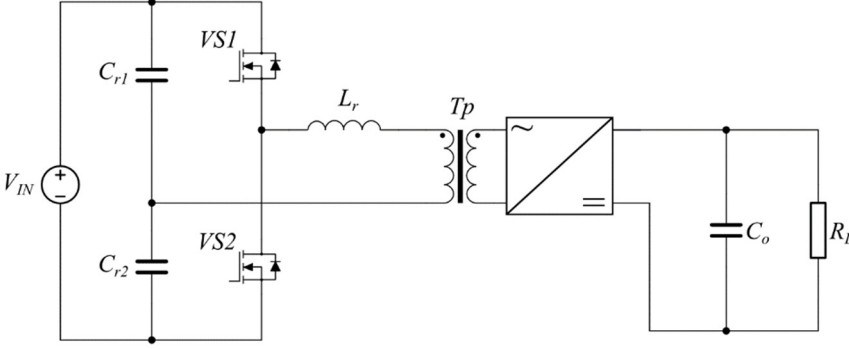

**Figure 1.** LLC DC-DC resonant converter with split capacitor.

The principle operation of the circuit in Figure 1 is the following: With the successive conduction of the transistors VS1 and VS2, the current flows through the primary coil of the transformer with a shape close to sinusoid. Then, the current is transferred to the secondary side where it is straightened from the rectifier and smoothed by the filter capacitor. This circuit is usually used for power converters up to 500 W [1,13].

If necessary, higher power uses a bridged or half-bridged type resonance converter. It is well known that the basic equations for each of the schematic variants considered are identical, differing only with the numerical coefficients described in [14–16].

When creating a mathematical model of the converter, the Kirchhoff laws are used, assuming that the input voltage $V_{in}$ is switched to the resonant circuit by means of the semiconductor switches VS1 and VS2. Thus, two major intervals, conventionally called positive and negative, are considered in the circuit. To determine the intervals of conduction for the transistors and their respective built-in reverse diodes, the following logic is used: If the direction of the current in the resonant circuit coincides with the polarity of the supply source, conduct the transistors—otherwise, the inverse diodes conduct the current. It is known from the operating principle of these circuits that energy is consumed when the transistors carry the energy from the power supply, and the return diodes return energy [7,8]. In the circuit diagram shown in Figure 1 there are three reactive elements, from which it follows that the system of linear differential equations is of the third order, having for each one of these reactive elements one equation of the state.

Thus, the equation system describing an LLC resonance transducer has the following form [13,14]:

$$\begin{pmatrix} L_1 + L_r & L_m \\ L_m & L_2 \end{pmatrix} \begin{pmatrix} \frac{di_1}{dt} \\ \frac{di_2}{dt} \end{pmatrix} + \begin{pmatrix} R_1 & R_m \\ R_m & R_2 \end{pmatrix} \begin{pmatrix} i_1 \\ i_2 \end{pmatrix} = \begin{pmatrix} fsw(t) \cdot U_{in} - u_{Cr} \\ -sign(i_2)u_{C0} \end{pmatrix}$$
$$C_r \frac{du_{Cr}}{dt} = i_1$$
$$C_0 \frac{du_{C0}}{dt} + \frac{u_{C0}}{R} = |i_2|$$

(1)

where $f_{sw}(t)$ is the switching function, which could be expressed as sign(sin($\omega$t)); $i_1$, $L_1$, $R_1$ are respectively the current, inductance and resistance of the primary winding, respectively, and $i_2$, $L_2$, $R_2$ are the parameters of the secondary windings; $L_m$ and $R_m$ are the mutual inductance and mutual resistance; $U_{in}$ is the supply voltage; $u_{Cr}$ is the resonant capacitor voltage; $u_{C0}$ is the output voltage; $C_r$ is the resonant capacitor; and $L_r$ is the resonant inductance.

Composition of the separate conduction intervals of the circuit operation is done via a switching function. This is a useful tool for modeling circuits with varying structures, such as power electronic converters [16].

*2.2. Verification of the Model*

Verification of the model was performed on a laboratory bench. The following TMDSHVRESLLCKIT LLC converter had a nominal power of 300 W, an input voltage from 375 VDC to 405 VDC, an output voltage of 12 VDC, a switching frequency from 80 to 150 kHz and an efficiency of more than 90%. A detailed description of the device is given in [17,18].

The values of the circuit elements are as follows: resonant Inductor (55 μH); resonant capacitor (24 nF); output capacitor (1520 μF); inductance of primary coil (285 μH); inductance of the secondary winding (0.97 μH); primary coil resistance (210 mΩ); resistance of the secondary coil (3.5 mΩ). The simulation results compared with the experimental data for several control frequency values are given in Figure 2. On Figure 3 are shown the results from the conducted simulations with the model by varying the output current of the converter.

Since the circuit is symmetrical, this is reflected by the switching function values of +1 and −1.

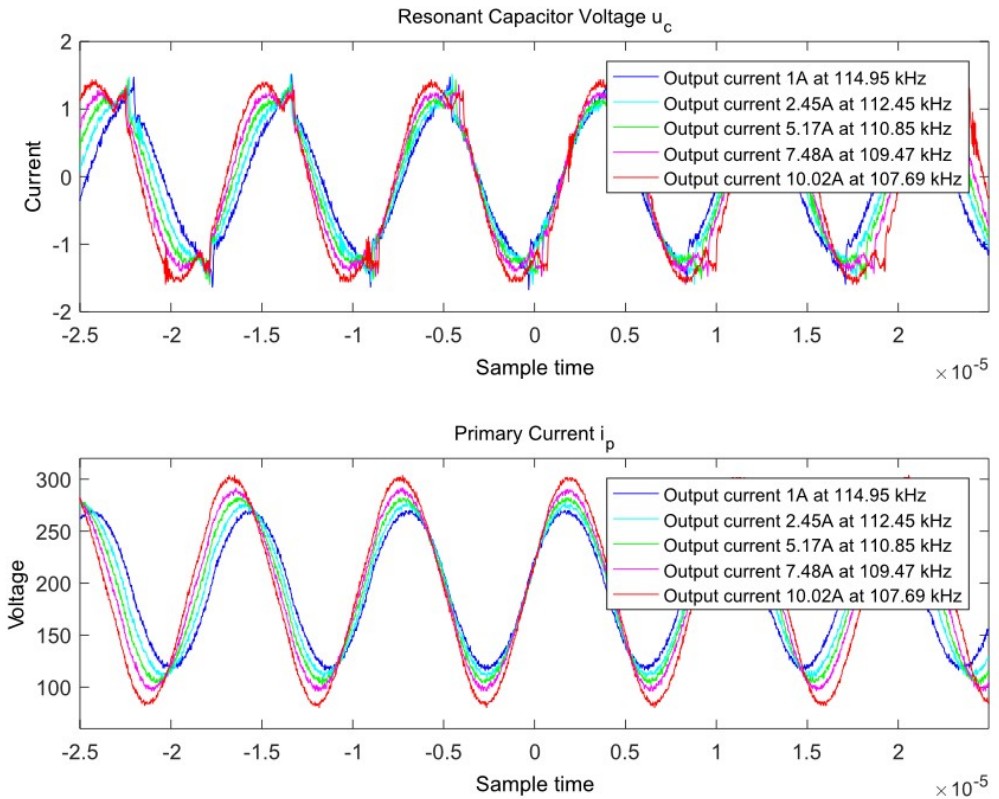

**Figure 2.** Measurements at five operating regimes, collected with a digital signal oscilloscope.

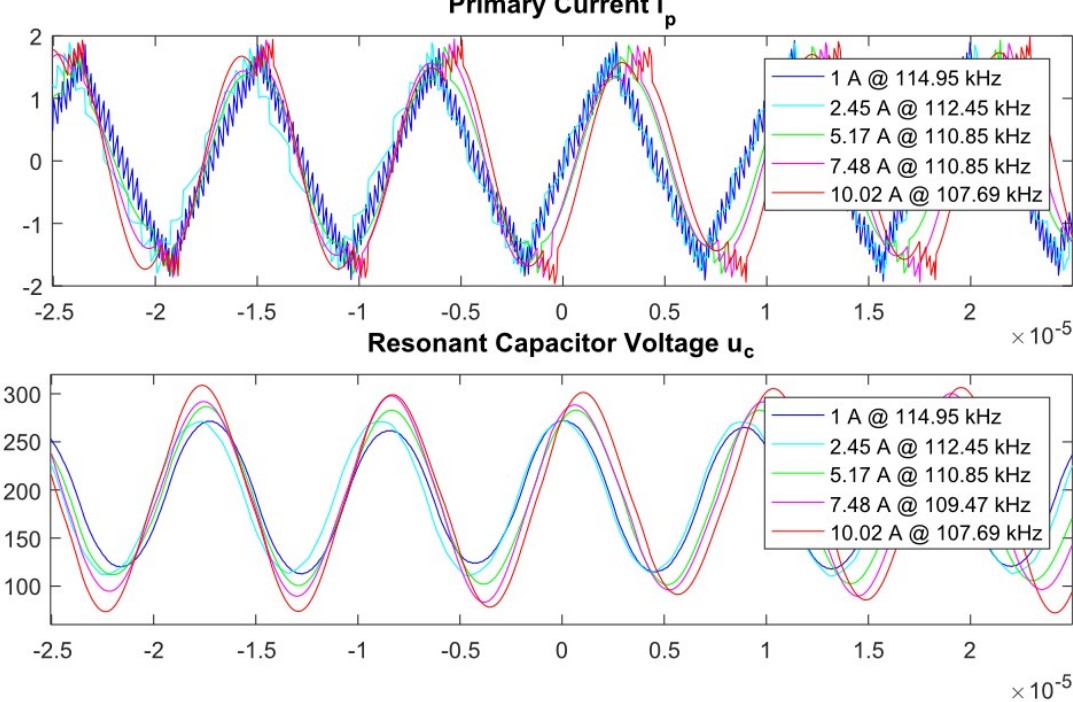

**Figure 3.** Waveforms of conducted simulations with the mathematical model.

### 2.3. Tolerance Analysis

It is known that no electronic element cannot be reproduced in practice with absolute precision, and, furthermore, that its characteristics change during operation as a result of changing its operating

temperature, aging and operating modes (current load level and voltage). Moreover, virtually every real element has some extent non-linear properties—that is, its characteristics are influenced by the stresses on it and the currents flowing through it. In this sense, guaranteeing the performance of the entire transducer is a task of guaranteeing the output parameters, regardless of the variation in the characteristics of the circuit elements.

The task of guaranteeing the indicators can be formulated as follows. In the case of setting up areas of variation for the circuit element characteristics, areas for changing the parameters and operating modes of the device are obtained. For example, with resonant inductance tolerances within ±20% of the nominal value and varying the resonant capacitance value, the effects of output voltage, resonant current and voltage on the resonant capacitor are investigated. Thus, an area of the operating modes of the resonant converter is defined that ensures it will operate within any range of output parameters.

### 2.4. Optimization Procedures

Optimization is a purposeful activity for obtaining the best results from a certain scenario, with certain limitations. In electronic transducers, these are most often: minimal loss, the critical-aperiodic transition process, realization of certain dynamics, appropriate modes of operation and so on. Criterion $I(x)$ is an assessment of the quality of the system in the form of a number that depends on the choice of certain variables $x$.

The formula for the optimization task is the following: limiting the starting current in the load by optimizing the selection of the $C_0$ filter capacitor and simultaneously preventing large voltage pulsations in the established mode. In this case, the criterion is the form of the load voltage, and the optimization variable is $C_0$. For this purpose, a suitable reference trajectory is selected $u_{C0ref}$. The selected path will be used to search for a suitable value for the filter capacitor so that the difference in values between the reference shape and output voltage is minimal. To do this, the optimization function is minimized [19,20]:

$$I(C_0) = \int_0^{t_{end}} \left(u_{C0} - u_{C0ref}\right)^2 dt \rightarrow min \tag{2}$$

From a mathematical point of view, a trapezoid reference trajectory with a proper time constant selection is more appropriate for finding an optimal value, because the algorithm will look for a more "aggressive" minimum of the function, although it is physically impossible to match its trajectory exactly.

Four equivalence constraints, which are the differential equation system describing the operation of the resonant converter Equation (1) and two constraint types of inequality, form a search area for a value of the output filter capacitor $C_0$:

$$\begin{aligned} C_{min} - C_0 \leq 0 \\ C_0 - C_{max} \leq 0 \end{aligned} \tag{3}$$

The task thus formulated is a $C_0$ value that satisfies the criteria of Equations (2) and (3) to satisfy the minimum requirement of the optimization function Equation (2). The systematically recorded task is limited to

$$\begin{aligned} &\min_{C_f} I(C_0) \; with \; constraints: \\ &\left| \begin{aligned} G_n(C_0) &= 0, \; for \; n = 1, 2, 3, 4 \\ G_n(C_0) &\leq 0, \; for \; n = 5, 6 \end{aligned} \right. \end{aligned} \tag{4}$$

This problem can be simplified using sequential quadratic programming [19–21].

In the sequential quadratic programming of each iteration step $k$, the function $I(x = C_0)$ and the limitations $G_n(x = C_0)$ are approximated by quadratic and linear functions, respectively. Thus, the task becomes the following:

$$\min_{d \in R^3} \nabla I(x_k)^T d + \tfrac{1}{2} d^T \nabla^2 I(x_k) d = F_k(d)$$
$$\nabla G_n(x) d = G_n(x_k), \; for \; n = 1, 2, 3, 4 \tag{5}$$
$$\nabla G_n(x) d \leq G_n(x_k), \; for \; n = 5, 6$$

where $\nabla I$ and $\nabla^2 I$ are the gradient and hessian of $I$, respectively, and $d$ is the direction of descent. If only one of the non-inequality constraints is inactive, they are turned off. After shutting down inactive restrictions, the task becomes the following:

$$\min_{d \in R^3} \nabla I(x_k)^T d + \tfrac{1}{2} d^T \nabla^2 I(x_k) d = F_k(d)$$
$$\nabla G_n(x)^T d = G_n(x_k), \; for \; n = 1, \ldots, l \tag{6}$$

This problem on the other hand is reduced to the problem of the unconditional optimization of

$$\min_{d \in R^3} \nabla I(x_k)^T Z_k d + \frac{1}{2} d^T Z_k^T \nabla^2 I(x_k) d = F_k(Z_k d) \tag{7}$$

where $Z_k$ is the projection matrix in which $\nabla G_n(x_k) Z_k = 0$, by $n = 1, \ldots, l$.

The unconditional optimization task is solved using a Newton or Quasi-Newton algorithm (Broyden–Fletcher–Goldfarb–Shanno algorithm). After determining the descent direction, the length of step $d_k$ is determined and the new iteration is calculated:

$$x_{k+1} = x_k + \alpha_k d_k \tag{8}$$

The step size $\alpha_k$ is calculated by minimizing the appropriate cost function, and the increase of $\alpha k$ stops if any inactive restriction changes into active.

The final unconditional optimization task is solved numerically by the following method: First, an initial value of the optimized magnitude $C_0$ falling within the acceptable range is selected. The system equations are solved, and the voltage $u_{C0}$ is removed in one period. The target function $I(x = C_0)$, descent direction for $I$ and length of the step are calculated. Correct $I$ by following the iterations of $C_0$ to remain within the permissible range. If the minimum is reached, the procedure stops; if not. the procedure is repeated.

## 3. Results

This section presents the results of the tolerance analysis of the transducer by comparing the output voltage and the inverter current in cases of unoptimized and optimized parameters.

### 3.1. Tolerance Analyses

Initially, dependence of the change in the resonance element values in an unoptimized circuit is determined (i.e., one that has been designed according to the known design methodologies). Let the tolerances of the circuit elements be ±20% of the nominal value for $L_r$ and ±10% of the nominal value for $C_r$. Figure 4 gives an example of nine combinations of resonant element values and their effect on the mean value of the output voltage.

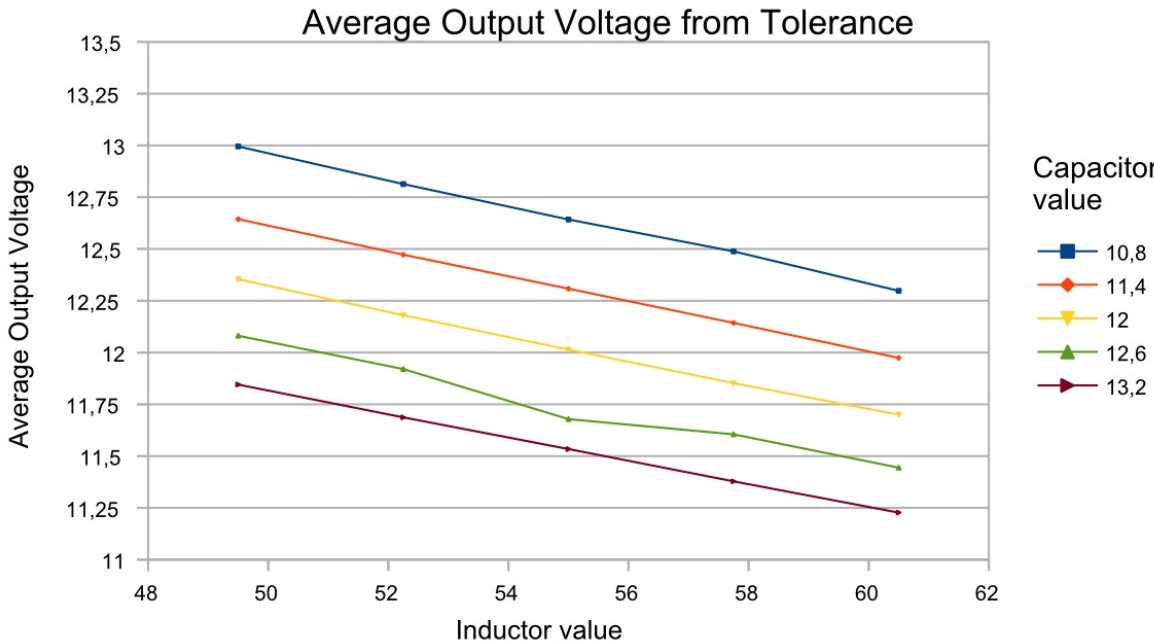

**Figure 4.** Tolerance analysis of the average output voltage.

Figure 5 illustrates the average pulsation of the output voltage at the same combinations of resonant circuit elements.

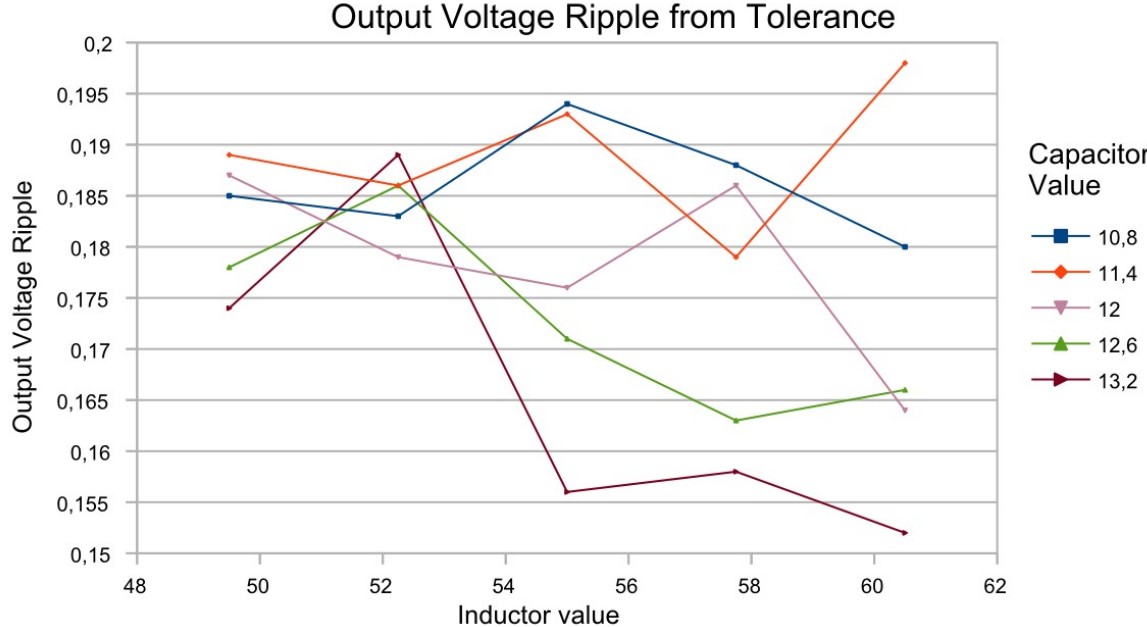

**Figure 5.** Tolerance analysis of the output voltage ripple.

With this tolerance analysis, it is determined that at element tolerances of ±20% and ±10%, respectively, there is a change in the output voltage range from 11.25 V to 13 V, as well as from 150 to 200 mV for the pulses.

In addition to the above-mentioned examples of output voltage and its pulsations, the other important parameters that are dependent on changes in the resonant element values and should be monitored are the resonant current and the voltage across the resonant capacitor.

The variation of the resonant current according to the tolerances is presented in Figure 6.

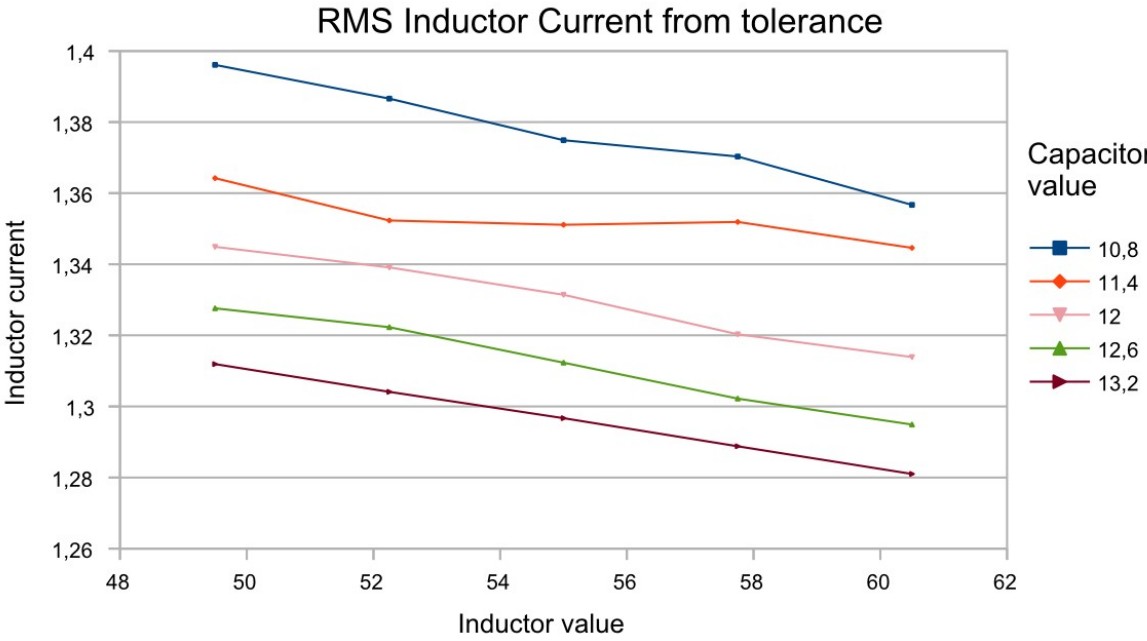

**Figure 6.** Tolerance analysis of the resonant current.

The resonant current directly influences the current through the semiconductor elements, so its reading is important in their selection. In addition, resonant inductor tolerances also affect the ratio of inductances $L_n$, which in turn alters the shape of the adjustment characteristic.

The voltage of the resonant capacitor is also influenced by the tolerances of the elements. A tolerance analysis against it is presented in Figure 7.

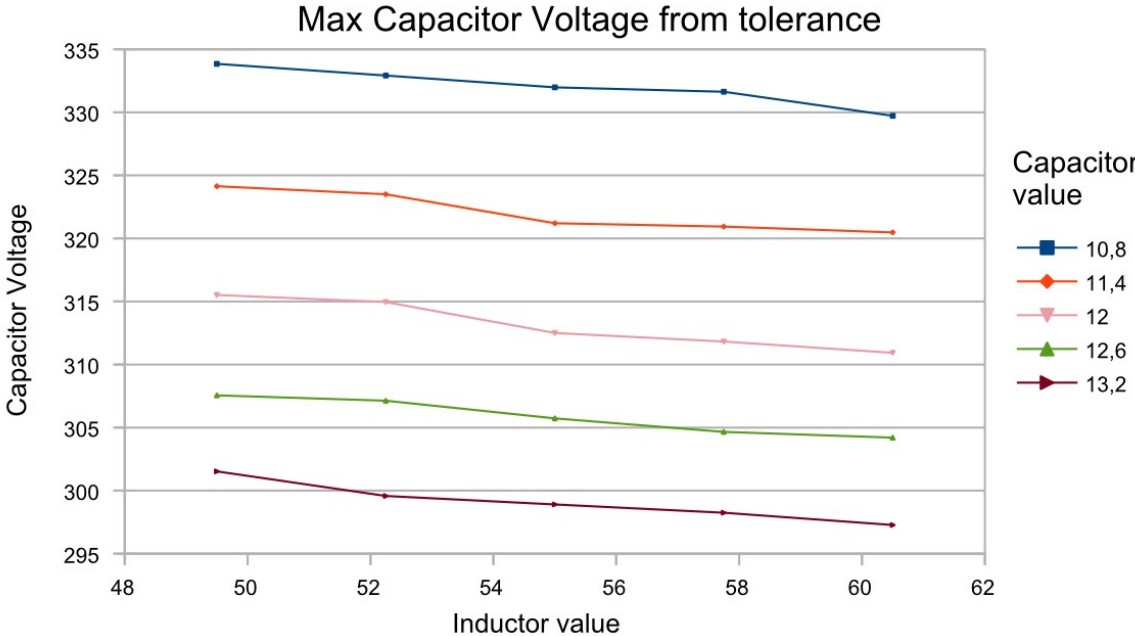

**Figure 7.** Tolerance analysis of the maximum capacitor voltage.

It can be seen from Figure 7 that the resonant inductor variation does not have a significant effect on the maximum resonance capacitor voltage compared with the resonant capacitance variation. The change in voltage due to a change in inductance is within 5 V, while the variation due to the capacitance is about 10 V for every 5% of its tolerance.

Element tolerances influence the shape of transient processes in the output. This is illustrated in Figure 8.

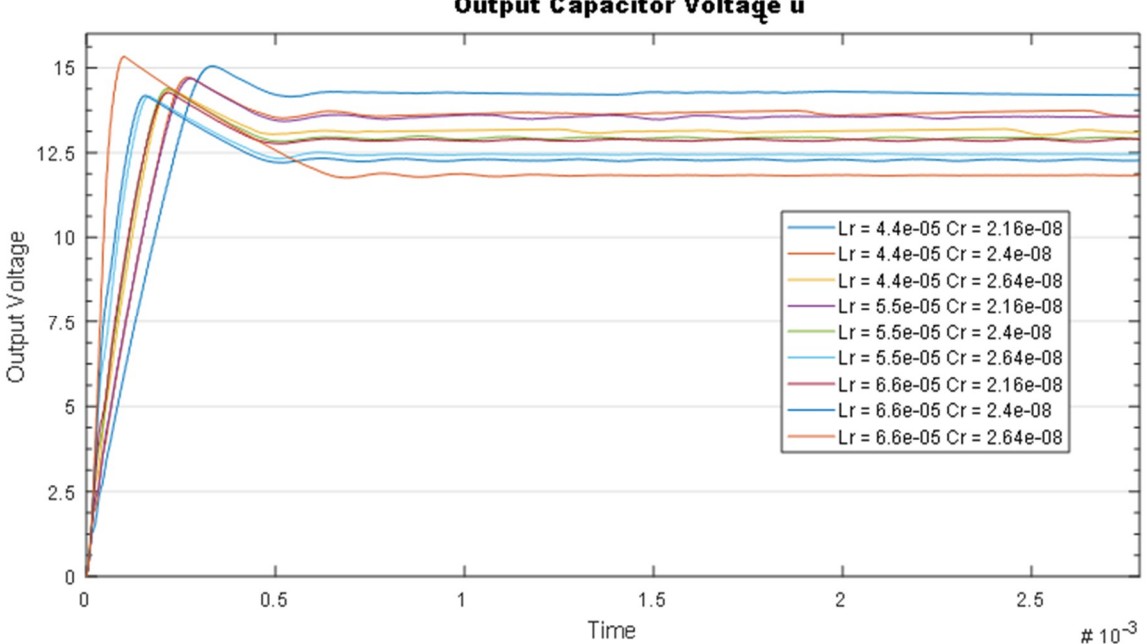

**Figure 8.** Tolerance analysis of the transient response of the output voltage.

As a result of the analysis of the results, it is found that in this case the shape of the output voltage is significantly influenced by the tolerances of the resonant elements. On the other hand, the tolerances of the elements considerably influence the transient process in the inverter circuit, as seen in Figure 9.

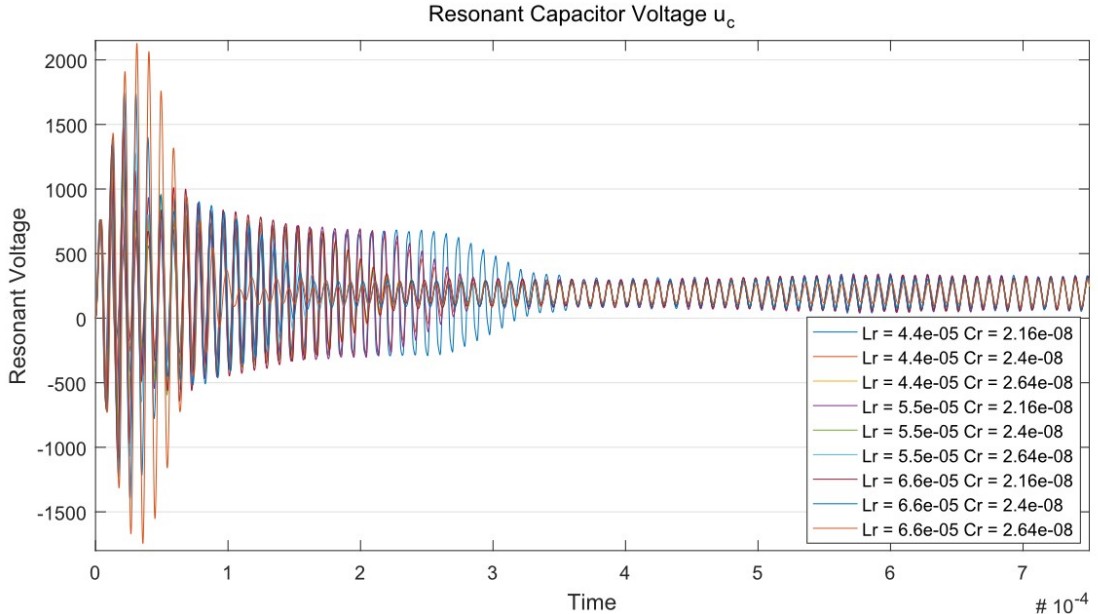

**Figure 9.** Tolerance analysis of the transient response of the resonant capacitor voltage.

In some cases, the voltage across the resonant capacitor can reach up to 195% of the nominal one, which will cause it to fail. In addition, the speed of the transition processes changes over time.

Table 1 shows the influence of tolerances on a change in the qualitative factor. The resonance element values vary so that the quality factor is preserved. It is set in certain steps, with iterations taking the minimum, mean and maximum values of the results obtained.

**Table 1.** Results when changing the resonance chain quality factor.

| $\Delta U_{0min}$ | $\Delta U_{0ave}$ | $\Delta U_{0max}$ | $U_{0min}$ | $U_{0ave}$ | $U_{0max}$ | **Q** |
|---------|---------|---------|---------|---------|---------|-----|
| 0.2023 | 0.3098 | 0.4849 | 12.7597 | 15.4814 | 18.622 | 0.5 |
| 0.2023 | 0.3025 | 0.4849 | 12.4329 | 15.0241 | 18.622 | 0.6 |
| 0.1829 | 0.3079 | 0.4849 | 12.2097 | 14.6717 | 18.622 | 0.7 |
| 0.1829 | 0.3162 | 0.4849 | 12.0670 | 14.3933 | 18.622 | 0.8 |
| 0.1595 | 0.3178 | 0.4849 | 11.9441 | 14.1616 | 18.622 | 0.9 |
| 0.1595 | 0.3129 | 0.4849 | 11.8479 | 13.9659 | 18.622 | 1 |
| 0.1595 | 0.3129 | 0.5196 | 11.7675 | 13.8 | 18.622 | 1.1 |
| 0.1518 | 0.3076 | 0.5196 | 11.7108 | 13.6543 | 18.622 | 1.2 |

$\Delta U_{0min}$ $\Delta U_{0ave}$ $\Delta U_{0max}$ are the minimum, mean and maximum pulsations of the output voltage, respectively, while $U_{0min}$, $U_{0ave}$, $U_{0max}$ are the minimum, mean and maximum values of the output voltage, respectively. The last column presents the value of the Q factor.

### 3.2. Optimization Procedures

The optimization of the resonant electronic resonator's output parameters consists of finding values of the circuit elements so as to obtain a good balance between the value of the output pulses, the speed of the transition process, the magnitude of the set-up voltage and so on. In this sense, a value of the output filter capacitor should be selected so that the specified requirements are optimally met.

Figure 10 shows a set of timing diagrams for the output voltage at output capacitor values obtained by optimization procedures.

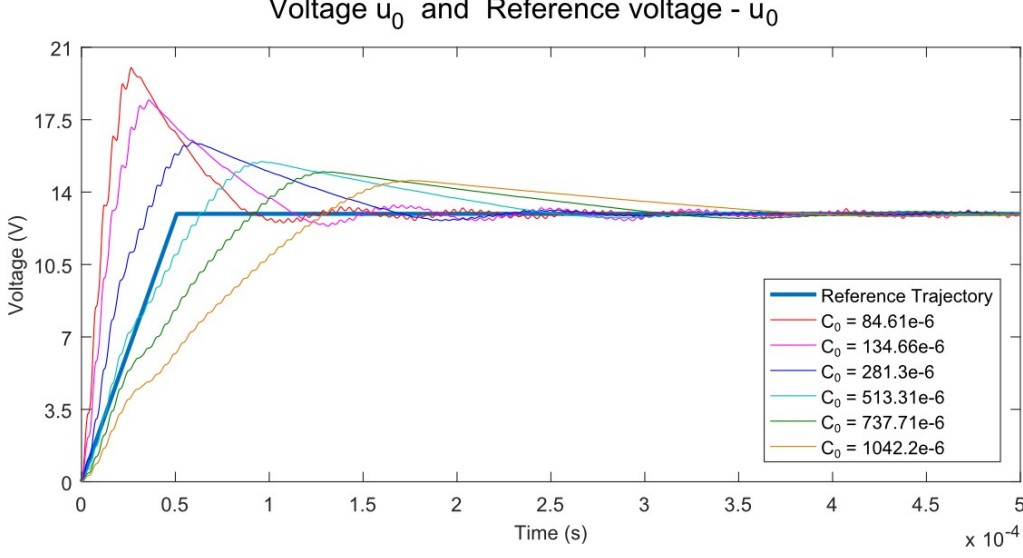

**Figure 10.** Optimization procedures for the output capacitor value.

The difference in choosing a range depends on the optimum value of the output capacitor that is being sought. The results obtained allow the scheme to function without a special soft start system, and a short duration of the transition processes is also observed. The result is that the resulting output voltage pulsation will increase with the lower output capacitor value. The designer's task is to select the optimal value search range to meet the requirements both in the transition process and in the established mode.

The choice of the lower limit of this range will depend on the maximum allowed pulses in the set mode, as well as the maximum-allowed rebound of the output voltage when it is set. The upper limit will depend on the maximum set-up time, as well as on the dynamics of the processes under sharp changes.

The optimization procedure looks for the value of resonant elements whose values satisfy a minimum deviation from a reference trajectory. The reference path is the shape of the resonant current or voltage transient process. In principle, it is irrelevant which of the two parameters the resonant elements are optimized for when looking for a particular shape of the transition process. The reason for this is the principle of the circuit action; it is known that the voltage on the resonant capacitor is proportional to the resonant current, so the shape of the transition processes is similar, the difference being only in the order of the values. When looking for a certain amplitude value, this output parameter is selected, which should be smaller when the optimal values are found. For example, if the optimal values for $L_r$ and $C_r$ are found where the resonant voltage does not exceed 600 V and the resonant current reaches 20 A, at the desired maximum value of 15 A a re-optimization is attempted to find optimal resonant current less than 15 A.

Figure 11 displays the result of the resonant current optimization procedure.

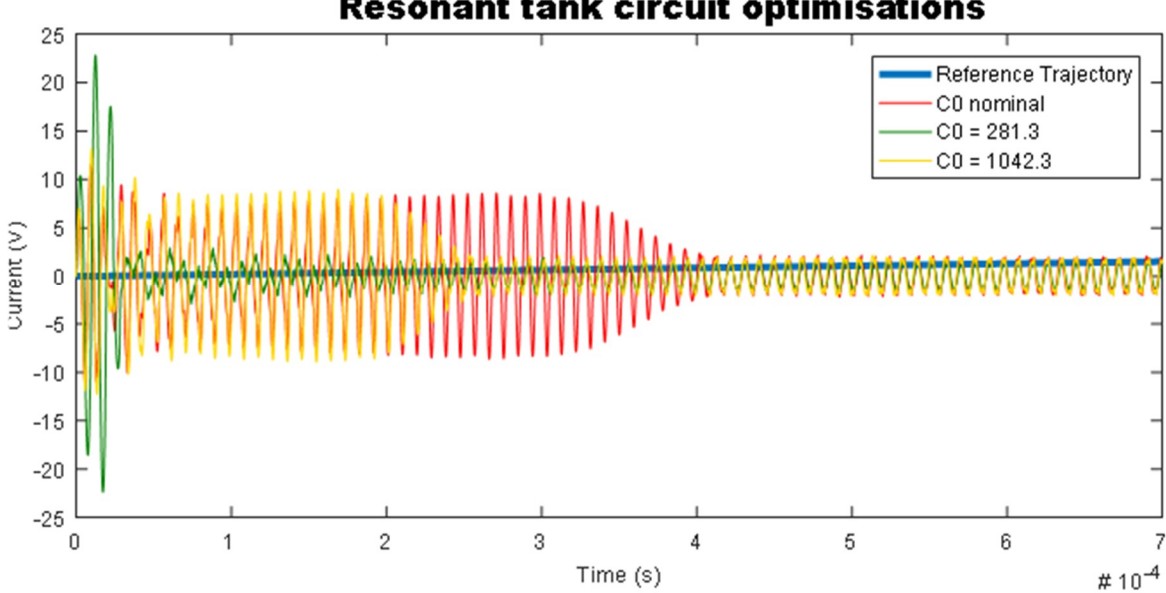

**Figure 11.** Optimization procedures of the resonant tank circuit elements.

In the present case, the optimal values were sought for resonant inductance and within the boundaries (minimum 12 nF, initial 24 nF, final 47 nF) for (minimum 25 μH, initial 55 μH, final 100 μH) the resonant capacitor at a reference path. The output capacitor had its nominal value. Additionally, a comparison of the transition process was given at three values of the output value—a nominal one, an optimum found with the lowest rebound of the output voltage and a compromise between the fast transition process and the least deviation from the reference path.

The values obtained for the resonant elements are as follows:

For output capacitor $C_0$ = 281.3 μF: $L_r$ = 90.2 μH, $C_r$ = 47 nF
For output capacitor $C_0$ = 1042.3 μF: $L_r$ = 57.91 μH, $C_r$ = 20.34 nF
For output capacitor $C_0$ = 1520 μF: $L_r$ = 70.16 μH, $C_r$ = 19.24 nF

Obviously the values of the obtained elements are highly dependent on the range in which the determined optimization is sought.

### 3.3. Tolerance Analyses with Optimized Elements

It would be of interest to make a tolerance analysis with the obtained optimal values of the output capacitor and resonant elements.

Figure 12 presents a tolerance analysis of the output voltage pulses with the set values obtained from the optimization procedure.

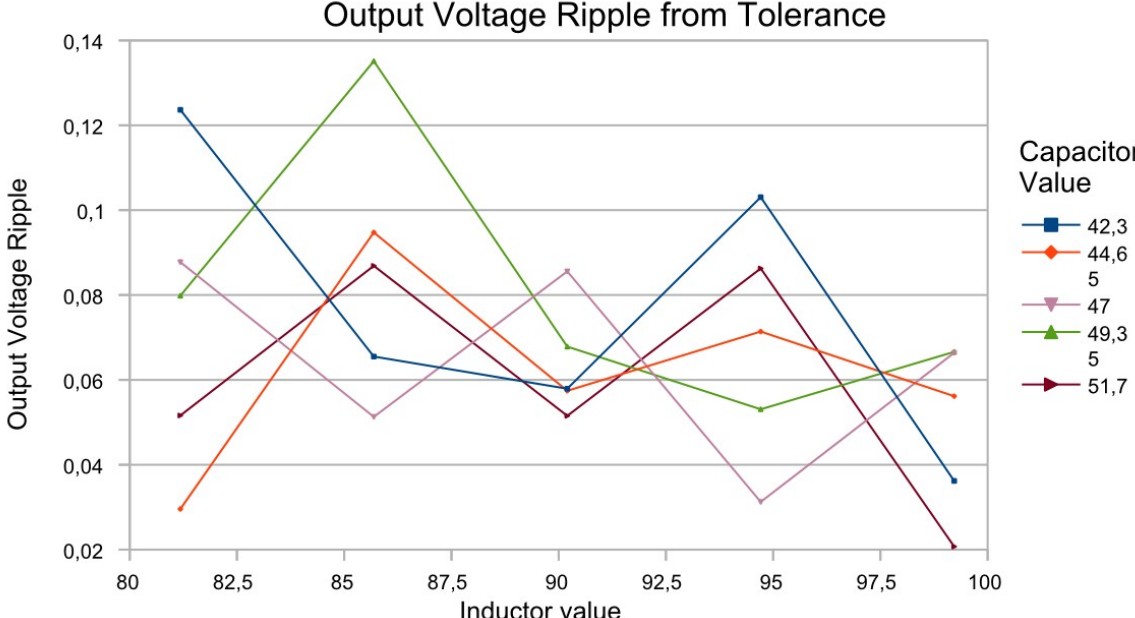

**Figure 12.** Tolerance analysis of the average output voltage with optimized values.

Referring to Figure 5, pulsations decreased by 40 mV, averaging between 60 mV and 80 mV, compared with 165 mV to 190 mV at nominal values.

Figure 13 presents a tolerance analysis of the mean value of the output voltage with the optimized values.

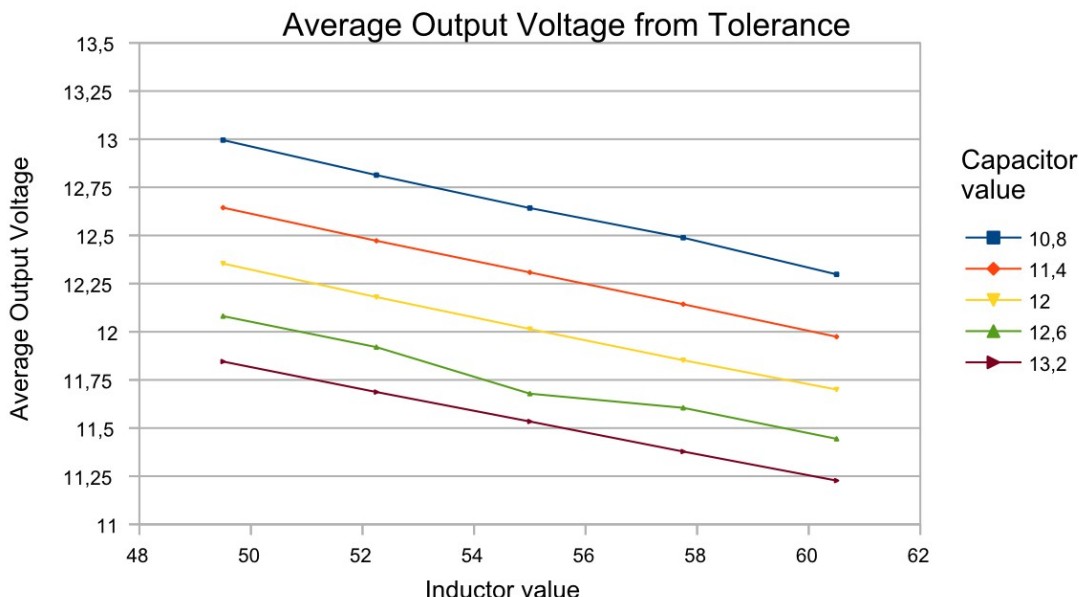

**Figure 13.** Tolerance analysis of the output voltage ripple with optimized values.

From a comparison with Figure 4, it can be concluded that the output voltage ranged from 12.1 V to 12.6 V at optimized values, and from 11.25 V to 13 V at the nominal values of the elements.

Figure 14 presents a tolerance analysis of the effective value of the resonant current with the optimized values.

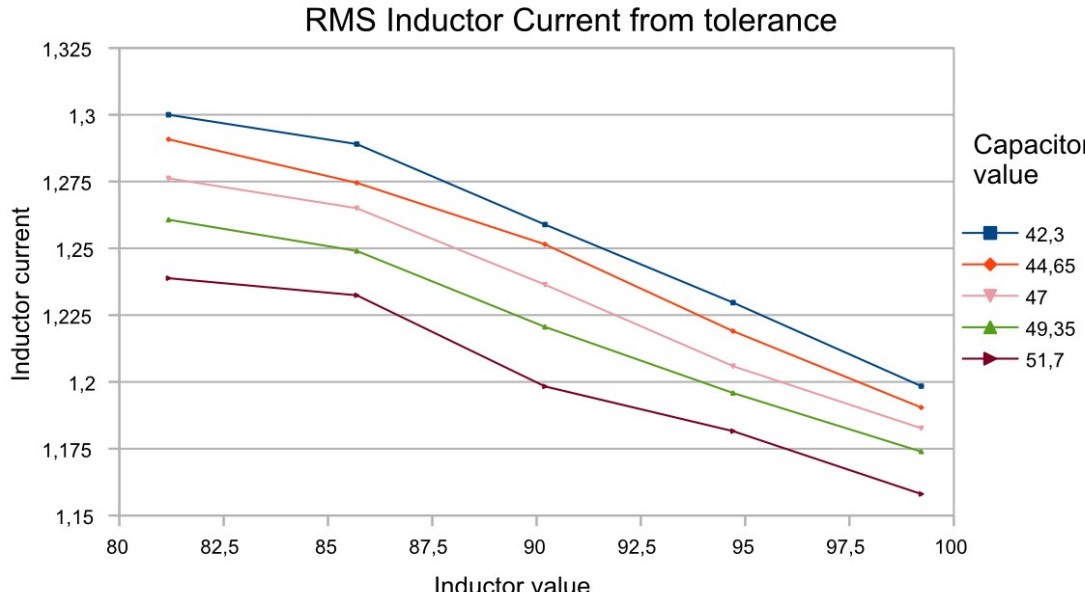

**Figure 14.** Tolerance analysis of the resonant current with optimized values.

Based on the results of Figure 6, the resonant current ranged from 1.3 A to 1.16 A at optimized values, and from 1.4 A to 1.28 A at the nominal values of the elements.

Figure 15 presents a tolerance analysis of the maximum voltage value on the resonant capacitor with the optimized values.

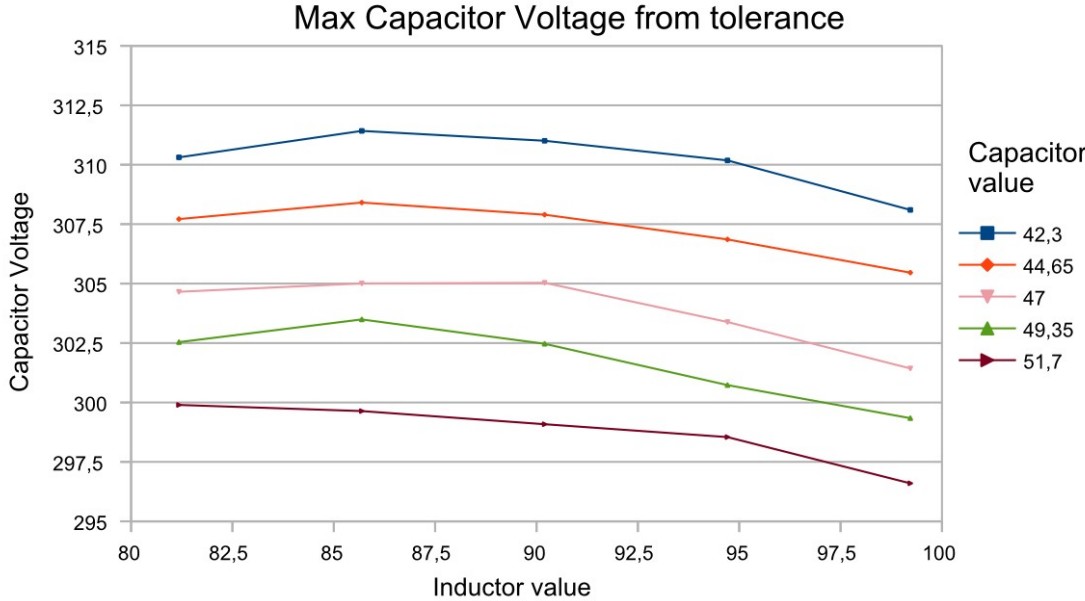

**Figure 15.** Tolerance analysis of the resonant capacitor voltage with optimized values.

As shown in Figure 7, the voltage was in the range of 295 V to 315 V at optimized values, and from 295 V to 335 V at the nominal values of the elements.

So far, the optimization balance is that the influence of the output quantities is less than the tolerances of the elements, and the operating modes have lower currents and voltages, which eases the operation of the electronic keys and leads to a decrease in losses and hence to an increase of efficiency.

Regarding the nature of the transient processes in the studied device, the following two figures represent differences in the optimized and non-optimized scheme. Figure 16 shows the determination of the output voltage, depending on the tolerances of the resonant elements.

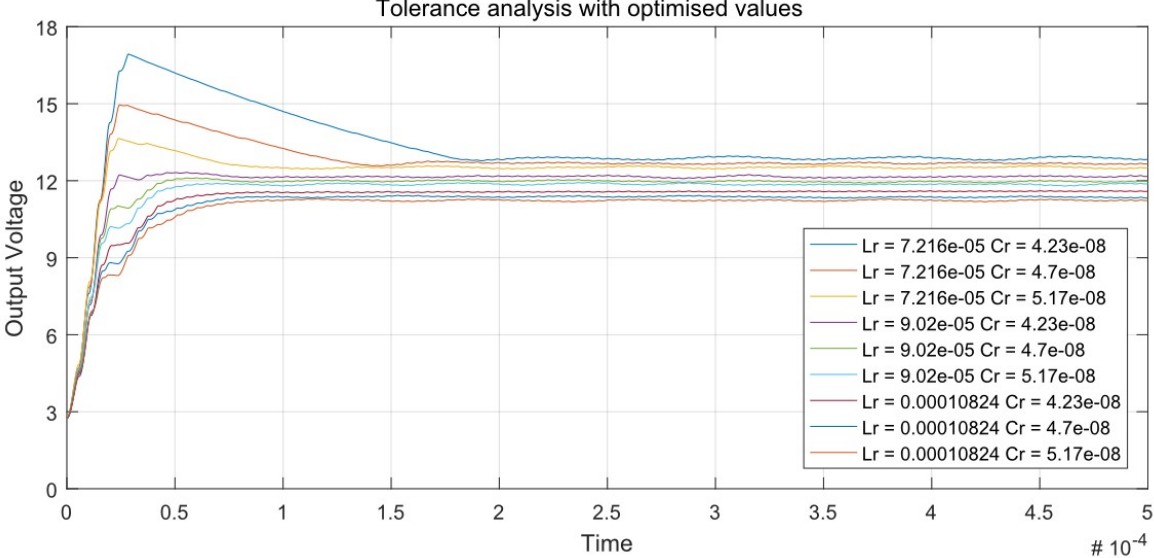

**Figure 16.** Tolerance analysis of the transient response of the output voltage with optimized values.

It is observed that in three out of six combinations, the transition process was aperiodical, whereas in the non-optimized tolerance analysis of Figure 8 there was none.

Figure 17 shows the transient voltage process on the resonant capacitor against the resonance tolerances.

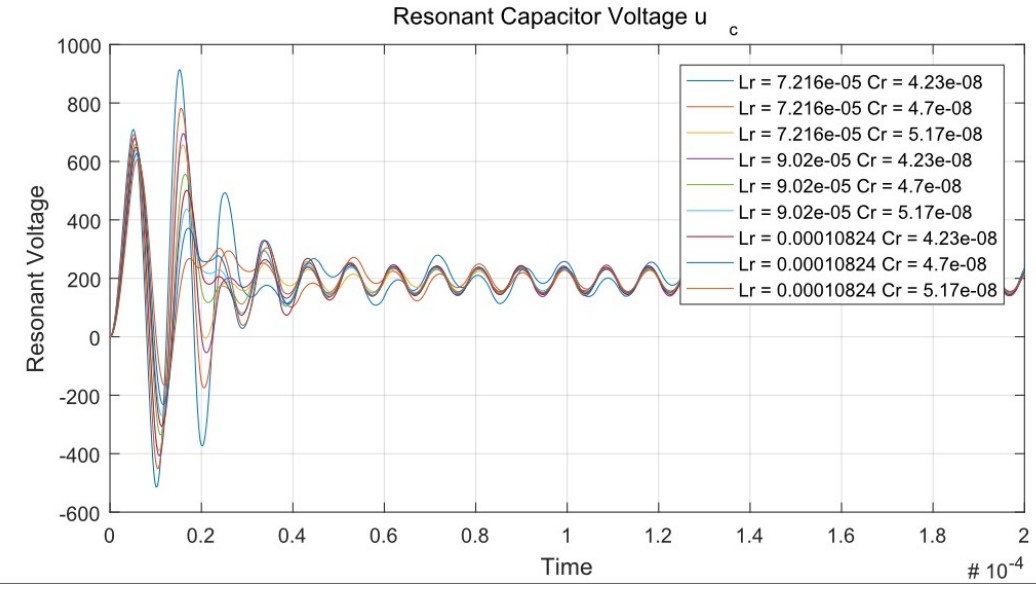

**Figure 17.** Tolerance analysis of the transient response of the resonant capacitor voltage with optimized values.

Compared to the transition process of Figure 9, it is noted that the transition process is relatively short—a maximum of three periods, where even in the worst case mode the peak voltage does not exceed 1 kV.

Another approach would be to select such values for the circuit elements so that the most commonly used operating modes meet the requirement for fast transition processes and, at the same time, small amplitudes of the selected quantities [22–24].

### 3.4. Tolerance Analyses with Optimized Elements using Other Optimization Procedures

In order to compare the results with the model-based optimization and prove its advantages, a comparison is made with the optimization of the same electronic converter, done with a standard tool in MATLAB/Simulink—Simulink Response Optimization [25]. The use of this tool is the closest to the idea that has been developed in the current work. Figure 18 shows the block diagram of the converter model described in this work and implemented in the MATLAB/Simulink environment. Additionally, the Simulink Response Optimization tool (in the green color of the figure) is added.

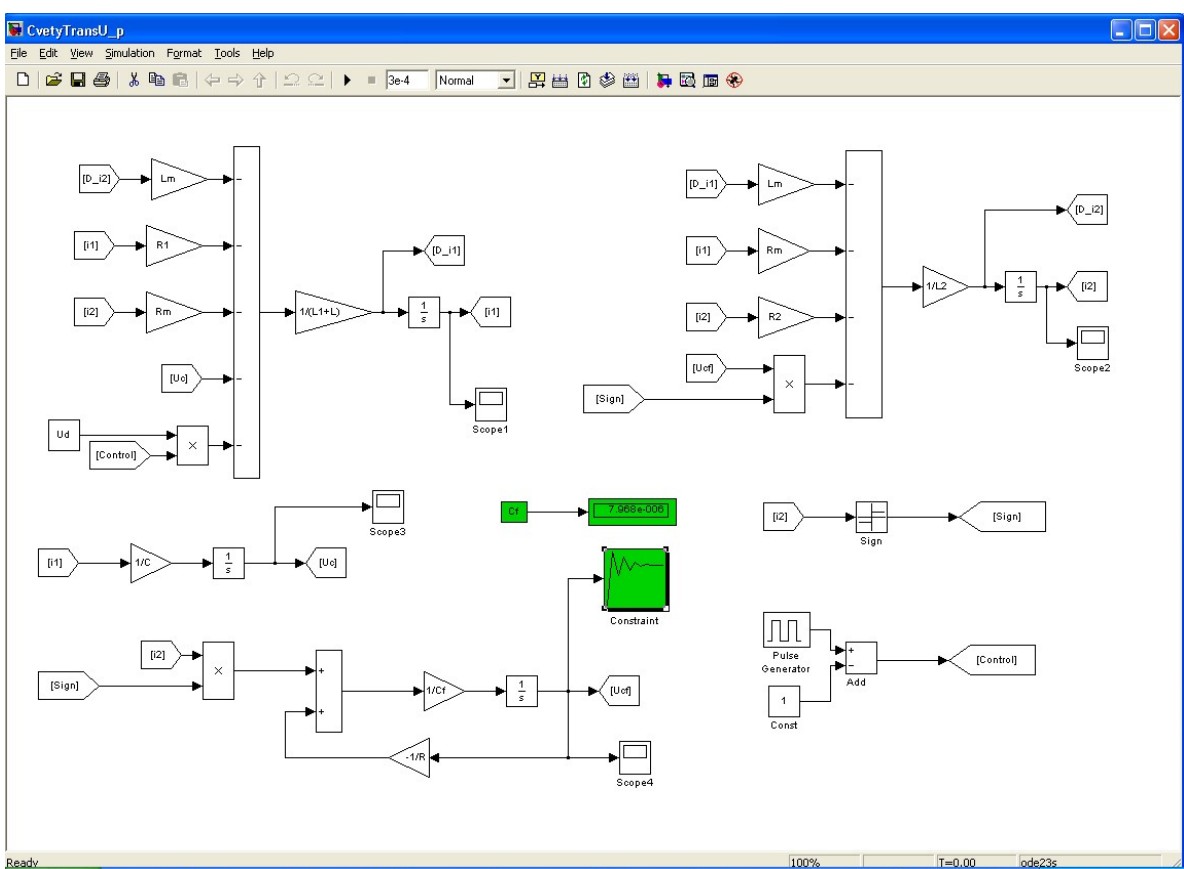

**Figure 18.** Block diagram of the converter model and the embedded Simulink Response Optimization tool.

With this tool, optimization is made on the output capacitor. The initial value for the output capacitor is given the value obtained at the design stage. Then, optimization parameters are set. As a result, the following value of the output capacitor is obtained: 7.6287 μF.

Figure 19 presents the results of a tolerance analysis at the obtained output capacitor value. From the comparison with those demonstrated with the model-based optimization in Figure 16, it is found that for those performed with Simulink Response Optimization the values obtained had a stronger dependence on the tolerance of the elements. This also shows the advantage of model-based optimization.

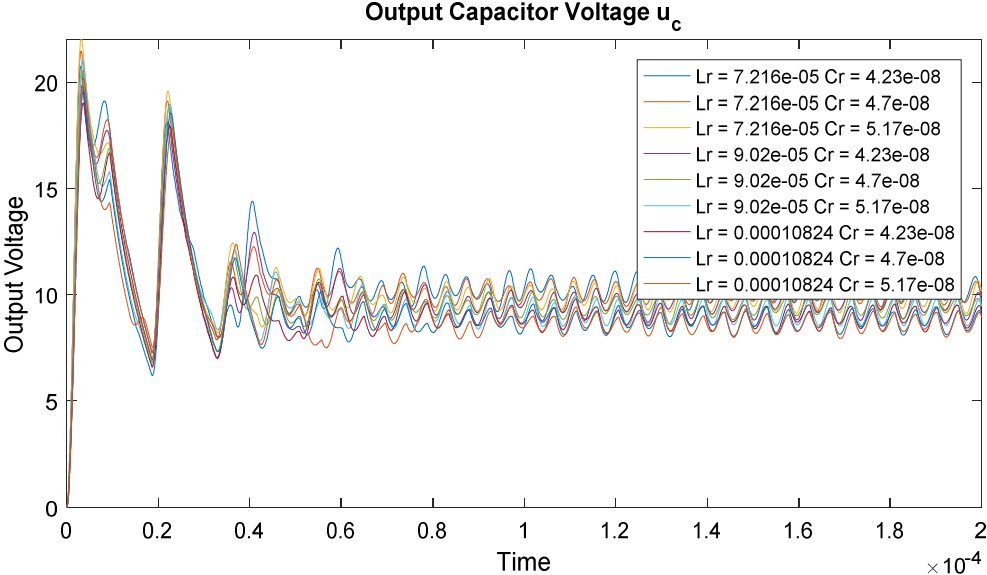

**Figure 19.** Tolerance analysis results with optimization, conducted with Simulink Response Optimization.

## 4. Conclusions

This paper proposes an approach for guaranteeing RLC converter parameters based on the use of computer-aided design technologies and mathematical modeling. Along with the algorithms of the mathematical methods, optimization procedures were performed according to different criteria. The advantages of this approach were proven using a tolerance analyses of optimized and non-optimized converters, and by using specific examples it was shown that the optimized converter possessed a series of advantages, including less dependence on circuit-element tolerances, shorter transient processes, better dynamics and lower maximum current and maximum voltage values on the device building blocks [26–28].

Tolerance analysis is a good approach to give the researcher or system designer an idea of the behavior of the circuitry values already calculated by the design stage, as well as what changes will occur in the operation modes that could be expected in the presence of certain percentages of tolerances in the physical elements. From then on, further design changes should be considered in order to determine more precise elements or more accurately adjust automatic electronic control systems.

Optimization procedures are a tool for finding the most appropriate circuit element values for already-known operating modes, as well as to relieve the operations of soft start systems and adjust the coefficients and thresholds of automatic control systems.

The conducted comparison with the standard optimization tool Simulink Response Optimization showed the advantage of the proposed approach by proving that dependence of the parameters of the converter output was 7% lower using model-based optimization.

The achieved results are important in that design methodologies for power electronic devices are based on an analysis in a steady-state and do not take into account dynamics and transient processes. By using model-based optimization, optimum values of the circuit elements are obtained, thus ensuring the output parameters are guaranteed when the values of the circuit elements vary.

The obtained results show that, when using the approaches together, a more robust device that is unsusceptible to circuit tolerances is assembled, with improved dynamics, faster transition processes, less reserve on circuit element values and less stress on the performance of the system for protection and automatic regulation.

**Author Contributions:** N.H., B.G. and T.H. were involved in the full process of producing this paper including conceptualization, methodology, modelling, validation, visualization, and preparing the manuscript.

**Funding:** The carried out research is realized in the frames of the project "Model based design of power electronic devices with guaranteed parameters", ДН07/06/15.12.2016, Bulgarian National Scientific Fund.

**Conflicts of Interest:** The authors declare no conflict of interest.

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
