# Peer review of "Model-Based Optimization of an LLC-Resonant DC-DC Converter"

_electronics, doi:10.3390/electronics8070799_

Round 1

Reviewer 1 Report

The authors responded to my comments. In my opinion the paper can be accepted for publication.

Author Response

Thank you for your review and the expressed opinion, that the paper is suitable for publication in Electronics.

Reviewer 2 Report

The paper has been improved from the previous comments. Nevertheless, there are some issues.

This paper still lacks scientific contributions. In the proposed model-based optimization procedure, the techniques to minimize the cost function with constraining and numerical methods are used, but these are basic theory and do not provide scientific contributions. Even if previous research has not been conducted, not all researches are worthy of journal articles. The scientific contribution of this paper should be clearly presented through sufficient research background and previous research survey. Therefore, the authors should add a review of previous studies to improve the dynamics and guarantee the devices' performance by optimization of parameters and explain the differences between conventional optimization methods and the proposed optimization methods in detail.

In addition, the answer to question 3 of reviewer 2 is unsatisfied. The authors should provide experimental results comparing the results of the conventional optimization methods with those of the proposed optimization method. In addition, a detailed analysis of the experimental results should be included.

Author Response

Thank you for your time in handling the review of our manuscript “Model-based Optimization of LLC Resonant DC-DC Converter " (electronics-535272).

We have revised our manuscript taking into account the review comments from the peer referees.  The revised version is sent to the editorial system. 

A detailed response to the comments is provided below.

The paper has been improved from the previous comments. Nevertheless, there are some issues.

Remark:

This paper still lacks scientific contributions. In the proposed model-based optimization procedure, the techniques to minimize the cost function with constraining and numerical methods are used, but these are basic theory and do not provide scientific contributions. Even if previous research has not been conducted, not all researches are worthy of journal articles. The scientific contribution of this paper should be clearly presented through sufficient research background and previous research survey. Therefore, the authors should add a review of previous studies to improve the dynamics and guarantee the devices' performance by optimization of parameters and explain the differences between conventional optimization methods and the proposed optimization methods in detail.

Answer:

It is not known to us the use of similar techniques to optimize the dynamics of power electronic devices – based on a verified model. If we had put in the title or abstract that we are proposing a new approach or method to guarantee the performance of electronic converters - that would sound a little bragging. The known optimization procedures are mainly aimed at minimizing the losses in the circuit elements. To improve the dynamics, it is usually relied upon the synthesis of a controller, usage of more modern control algorithms such as neural networks, predictive control, and so on. This is also shown in the selected block of literature, dedicated to improving the dynamics of the power electronic converters:

Xuejun Ma, Xiumei Yue, Hongxia Wu, & Jinhua Liu. (2009). A novel method of improving the dynamic response of DC-DC converter. 2009 IEEE 6th International Power Electronics and Motion Control Conference. doi:10.1109/ipemc.2009.5157597

Kurokawa, F., & Sukita, S. (2007). A New Model Control DC-DC Converter to Improve Dynamic Characteristics. 2007 7th International Conference on Power Electronics and Drive Systems. doi:10.1109/peds.2007.4487789

The main contribution in our opinion is that we are going to make the most of the power circuit itself, and improved control method is another approach and an alternative to resolving issues related to dynamics.

Remark:

In addition, the answer to question 3 of reviewer 2 is unsatisfied. The authors should provide experimental results comparing the results of the conventional optimization methods with those of the proposed optimization method. In addition, a detailed analysis of the experimental results should be included.

Answer:

A new section has been added to the manuscript - 3.3. Tolerance analyses with optimized elements using other optimization procedures. In this way, a comparison of the proposed method of model based optimization with other methods is used. The results show the benefits of model-based optimization.

Round 2

Reviewer 2 Report

The paper has been improved from the previous comments. The minor revision is needed.

1.    The authors should add literatures provided in answer to reference. In addition, provide the limitations of the literature in the introduction.

2.    Figure 10 shows the output voltage on various output capacitors. In my opinion, Figure 16, rather than Figure 10, is appropriate to compare the proposed method with the conventional method.

Author Response

Dear Reviewer,

We kindly accept the constructive critic and we have submitted a final document with the improvements. As usual we are open to other issues regarding the paper.

Kind Regards,

Nikolay Hinov, Bogdan Gilev and Tsveti Hranov

This manuscript is a resubmission of an earlier submission. The following is a list of the peer review reports and author responses from that submission.

Round 1

Reviewer 1 Report

This paper discussed the model-based optimization of LLC resonant converter. The optimization procedure is easy to understand. In the introduction, however, this paper did not provide sufficient research background, and there is no review of previous studies. In addition, most of the scientific evidence presented is the result of previous studies and either there is no scientific contribution or contribution is not clear. This paper is similar to the application note for the design of LLC Resonant Converter[Extra ref1, 2]. The authors need to demonstrate logical analysis rather than an empirical approach by various experimental results. It is strongly recommended that the authors resubmit the paper with a mandatory revision including scientific contribution.

[Extra ref1]: AN-6104 LLC Resonant Converter Design using FAN7688, Fairchild Semiconductor Corp.

[Extra ref2]: AN 2012-09 Resonant LLC Converter: Operation and Design, Infineon Technologies North America Corp.

Reviewer 2 Report

The paper presents a model-based optimization to guarantee the performance of the LLC DC-DC converter. The primary scope of the study is to maintain the output parameters regardless of the variation of the values of the circuit elements.

The authors have to include experimental setup used to carry out Figs. 1, 2. Refs. [11] [12] are not sufficient to understand. The switching frequency varies between 107 -114 kHz, but the device works in 80-150 kHz, please explain the choice of the frequency range in experimental results.

The authors write: “The advantages of this approach have been proven to be a tolerance analysis of an optimized and non- optimized converter, and it has been shown with specific examples that the optimized has a series of advantages such as less tolerance, shorter transient process, better dynamics, and lower maximum current and voltage values in the device”.

Authors should include results on comparisons of the proposed optimization technique with others presented in the literature to better highlight these advantages.